# Resource Allocation for Downlink Full-Duplex Cooperative NOMA-Based Cellular System with Imperfect SI Cancellation and Underlaying D2D Communications

**DOI:** 10.3390/s21082768

**Published:** 2021-04-14

**Authors:** Asmaa Amer, Abdel-Mehsen Ahmad, Sahar Hoteit

**Affiliations:** 1School of Engineering, Lebanese International University, Al Khyiara, West Bekaa, Lebanon; abdelmehsen.ahmad@liu.edu.lb; 2Laboratoire des Signaux et Systèmes, Université Paris Saclay- CNRS -CentraleSupélec, 91190 Gif-sur-Yvette, France; sahar.hoteit@universite-paris-saclay.fr

**Keywords:** cooperation networks, device-to-device communication (D2D), full-duplex, matching theory, non-orthogonal multiple access (NOMA)

## Abstract

In this paper, the interplay between non-orthogonal multiple access (NOMA), device-to-device (D2D) communication, full-duplex (FD) technology, and cooperation networks is proposed, and a resource allocation problem is investigated. Specifically, a downlink FD cooperative NOMA-based cellular system with underlaying D2D communications is proposed, where, in each NOMA group, the strong user assists the weak user as an FD relay with imperfect self interference (SI) cancellation. In terms of reaping spectral efficiency benefits, the system sum rate is to be maximized by optimizing channel allocation. This optimization is based on quality of service (QoS) constraints of D2D pairs and cellular users (CUs), power budget of base station and strong user (cooperative phase), and successive interference cancellation (SIC) constraints. Since the maximization formulated problem is computationally challenging to be addressed, a two-sided stable many-to-one matching algorithm, based on Pareto improvement, performs sub-channel assignment. Extensive simulations are implemented to demonstrate the system performance indicated by different metrics.

## 1. Introduction

The explosive growth of internet of things (IoT) is revolutionizing today’s world and introducing advanced applications in all aspects. Consequently, the number of IoT devices connected to the wireless network is immensely growing in a blistering pace, so mobile data is witnessing a steep rise. To fulfill the requirements of IoT vision, the fifth generation (5G) and beyond are expected to realize an evolution in the present networks. So, for fitting the quality of service (QoS) guarantees of the massively connected devices, advanced 5G key technologies are considered. The present orthogonal multiple access (OMA) schemes assign different resources (frequency, time, code, or space) to different users, thus limiting the number of users to the number of scarce resources. These schemes cannot fully support the 5G and beyond demands; hence, more efficient radio multiple access (MA) schemes are needed [1].

Non-orthogonal multiple access (NOMA), in contrast to the previous OMA schemes characterizing the previous generation networks, is unique in its key design. It allows transmitter to serve multiple users with simultaneous sharing of the same resource blocks (RBs) via superposition coding (SC) of the signals. It differentiates them in the power domain by allocating them different power levels, and while allowing limited multiple access interference (MAI) at the receiver, successive interference cancellation (SIC) is performed there. Note that NOMA term is interpreted not only for the power domain but also for the code domain. So, an overloaded system having multiple users per available RB (i.e., available code) can be categorized as code domain NOMA-assisted system [1], and which is out of the focus of this work. NOMA has been recognized as trumping candidate over its counterpart OMA schemes to be empowering key for 5G and beyond networks, and to be adopted by standardization bodies. This owes to its ability to provide better massive connectivity, low latency and higher spectral efficiency [2], even with the presence of maximum MAI at receivers that is not degrading performance anymore due to the advancements in interference cancellation techniques [1].

Resource allocation is considered to be challenging in the coming complex dense network architectures. So, for reaping the benefits of introducing NOMA instead of OMA, several researches have been conducted for developing efficient resource (i.e., power, sub-channel, and computing resources) allocations, to optimize different system metrics (e.g., spectral efficiency, energy efficiency, latency, etc.) and cope with the introduced challenging interference. Authors in References [3,4,5,6,7] have studied sub-channels and power allocation in NOMA systems. In References [3,4,6], resource allocation in single-cell downlink NOMA-assisted cellular systems have been studied. Particularly, authors in Reference [3] aimed to find the suitable clustering of cellular users (CUs) over sub-channels under NOMA principle by using matching theory [8], formulating the clustering problem as two-sided many-to-one matching with externalities problem, and sub-channels and CUs are the sides of the two-sided beneficial matching game, each aiming to realize his own benefits (i.e., increase the achievable rate) and to reach the optimal system sum rate, as total system performance indicator. In contrast, in Reference [4], authors have formulated sub-channel allocation as a many-to-many matching algorithm, so that each sub-channel can be assigned to multiple users and each user can utilize multiple sub-channels, besides that, authors also studied power allocation between users sharing the same sub-channel. Given fixed channel allocation, in Reference [6], various optimal power allocation approaches have been investigated with 2-user NOMA, multi-user-NOMA and multi-channel-NOMA schemes. Moreover, considering both uplink and downlink NOMA transmission, in Reference [5], efficient user clustering and power allocation approaches have been developed, while Reference [7] has addressed the multi-cell NOMA systems and coped with additional interference between cells.

Besides conventional NOMA systems, cooperative NOMA was first introduced in Reference [9], to extend coverage and boost reception reliability. In this cooperative scenario, strong users can act as relays to assist weak users sharing the same sub-channel, by exploiting SC and SIC techniques due to the fact that the weak user’s signal already exists at the strong user. Thus, the performance of the weak user will be improved, while its QoS requirements are more guaranteed. Due to the fact that energy efficiency is also considered as 5G key objective, researchers have studied energy harvesting with simultaneous wireless power and information transfer (SWIPT) integrated with cooperative NOMA. Thus, energy harvesting motivates the strong user to be acting as a relay assisting the weak users without consuming its own battery [10]. It is worth noting that integrating cooperative networks with NOMA is not only considered in the previous scenario (cooperation between users) but also with using dedicated relays between base station (BS) and users for forwarding signals to the latter and working on best relay selection schemes analysis [11].

Moreover, in cooperative networks or relaying, different studies have limited it for half-duplex (HD) relaying, so source-relay and relay-destination channels are kept orthogonal. This leads to approximately 50% loss in spectral efficiency due to need of extra time slot or extra frequency. Therefore, full-duplex (FD) technology can be considered as an empowering key technology and particularly the in-band FD, which is known as the reception and transmission on the same frequency band and at the same time [12,13], although it was considered impossible during an earlier period [14]. So, due to higher spectral efficiency demands and scarcity of resources, in-band FD can be implemented at the relay node. However, the relay will experience self-interference that loops back to it. This interference is faced using SI cancellation techniques that have witnessed high advancements [15]. So, with existence of advancements in these cancellation techniques, even if no perfect cancellation is provided, FD technology benefits can be reaped to support the fast growth of wireless communication. So, with integration of cooperative NOMA and in-band FD, strong user (or the dedicated relay) can assist the weak user at the same frequency band simultaneously.

Another promising 5G key technology is device-to-device (D2D) communication, that allows the direct communication between devices without traversing the core network, thus avoiding overwhelm of the system. D2D is categorized into in-band and out-band based on whether the cellular spectrum is also used for D2D communication or not, respectively. It is worth it to say that the former will introduce interference between cellular tier and D2D tier, especially when same resources are shared between CUs and D2D pairs (i.e., underlaying D2D) [16]. So, investigating efficient resource allocation and interference management approaches for underlaying D2D communication was researchers interest because of its beneficial gains including reuse gain, hop gain, and proximity gain and its ability of cellular coverage extension [17].

## 2. Related Works

Several researches have studied the interplay between D2D and NOMA for extracting their holding-on-promise beneficial integration in terms of different system performance metrics, and under interference management methods [18,19,20]. In Reference [18], application of NOMA in D2D communication with the new concept of D2D groups (i.e., to serve multiple D2D receivers with one D2D transmitter) was introduced, where NOMA principle is applied to serve the same-D2D-group multiple receivers over same sub-channel, and D2D communication in each D2D group shares the sub-channel used by a single cellular user. Authors in Reference [18] aimed to increase the total system sum rate, and cope with intra-D2D group interference between receivers of same NOMA group, inter-D2D-group interference caused from another D2D group sharing same sub-channel and interference caused by the cellular user. They formulate sub-channels allocation problem as many-to-one matching problem where multiple D2D groups can share one sub-channel with only one cellular user, followed by optimized power allocation in each D2D group, under QoS, total transmit power, and SIC constraints. In contrast to References [18,19,20] have studied the application of NOMA between CUs, and not for D2D transmission. In Reference [19], authors have implemented D2D underlaying uplink cellular network, so multiple D2D pairs and multiple CUs can share one sub-channel. Trained by results of channel and power allocation solution based on sequential convex optimization, convolutional neural network algorithm(CNN) is implemented aiming to maximize the total achievable data rate of D2D pairs under QoS requirements of CUs. Authors in Reference [20] have also studied resource allocation for D2D underlaying NOMA-based cellular network aiming as [19] and under SIC requirements and CUs QoS constraints, but by allowing only one D2D pair to be operating on the sub-channel.

Full-duplex cooperation in NOMA-based cellular systems was addressed in References [21,22,23,24,25] with different cooperation scenarios. Authors in References [21,22,23] have considered the scenario of cooperation between users, where strong user will assist the weak user. On the other hand, authors in References [24,25] have studied the scenario of using dedicated relays between the source (i.e., BS) and the intended users. In Reference [21], authors studied a downlink cooperative NOMA system with two pre-paired NOMA users, where the strong one acts as a full-duplex relay assisting the weak user, and can harvest energy through power splitting SWIPT technology. The aim in Reference [21] was to maximize the energy efficiency under QoS constraints of the far user only, and the power budget constraints of the base station and the strong user. The work in Reference [22] has investigated a similar system to that in Reference [21] but without adopting SWIPT. Moreover, authors in Reference [22] have covered additional cases, i.e., relay (i.e., the strong NOMA user) switching between HD and FD mode, and having no direct link between BS and the weak user. A performance analysis is shown in Reference [22], in terms of outage probability (OP), ergodic rate, and energy efficiency for each covered case. In Reference [23], an in-band full-duplex cooperative NOMA system is studied with the generality of no direct link between BS and weak user, where OP and ergodic rate are analyzed, and a power allocation is formulated to minimize OP. Two dedicated-relays selection schemes in FD/HD cooperative NOMA system were addressed in Reference [24], so to study the difference in the impact of the two schemes on the outage probability. In Reference [25], imperfect SIC, along with in-phase and quadrature-phase imbalance is also considered while analyzing outage probability of NOMA users. Figure 1 shows the literature review. Their distribution over the 4 technologies is shown, where some of them have addressed resource allocation in their proposed scenarios, and others were only performance analysis works.

### Motivations and Contributions

To the extent of our knowledge and based on the aforementioned works, no existing work from the literature has studied resources (i.e., sub-channels) allocation in FD cooperative NOMA-based cellular system with imperfect SI cancellation and underlaying D2D communications. This is shown in Figure 2. In this work, we will consider it, so multiple CUs will be grouped to form a NOMA group over one sub-channel. In each NOMA group, the strong user will act as an FD relay with an imperfection level of SI cancellation, and it will assist the cell-edge user (weak user). Besides this, multiple D2D pairs will be allowed to share the sub-channel with each NOMA group. Sub-channel assignment will be optimized to maximize the system sum rate and enhance the performance of cell-edge users, thus extending the coverage and, consequently, increasing the number of assigned D2D pairs (Hereinafter, “number of assigned D2D pairs” will indicate to the number of D2D pairs that are accepted to be assigned to the sub-channels of NOMA groups; the acceptance depends on the interference level on NOMA group users, considering QoS requirements.). The major contributions of this work are summarized as follows:FD Cooperative NOMA technique is integrated with underlaying D2D, for reaping their beneficial integration in terms of better spectral utilization and better cell-edge users performance and that by methodical interference management. In this scenario, resource allocation problem must be solved efficiently. For this aim, sub-channels are assigned between NOMA groups and the D2D pairs.First, based on initial fixed power allocation coefficients, sub-channel assignment is solved via a many-to-one matching theory based on Pareto improvement. Meanwhile, QoS guarantees (quantified by received signal to interference and noise ratio (SINR) thresholds) for both CUs and D2D pairs, SIC success requirements in NOMA groups, power budgets, and SI level due to FD operation are taken into consideration.Second, both HD and FD modes operating at the strong user in each NOMA group are modeled and compared. This comparison studied the effect of SI cancellation level, received SINR thresholds, and the density of D2D tier (total number of D2D pairs) on the performance difference between applying both modes.

The rest of the paper is structured as follows. Section 3 describes the system and channel model. The optimization problem is formulated in Section 4, and the sub-channel allocation is presented in Section 5. Simulation results are given and analyzed in Section 6. Finally, the work is concluded in Section 7. Table 1 includes notations of the paper.

## 3. System Model

### 3.1. System Description

Consider the downlink transmission scenario of a single cell with one Macro-base station (MBS). The MBS serves a set of *U* CUs denoted by C, and operates on a set of K orthogonal sub-channels K = {1, 2, …, K }. The CUs, which are assigned over the same sub-channel *k*∈K, are grouped to form a NOMA group, with *m* users per NOMA group, where *m* ranges between 2 and |C| (|.| denotes the cardinality of a set). In each NOMA group, intra-group cooperation is applied, where the strong user with greater channel coefficient (i.e., near to the MBS), acts as a relay that can operate in FD mode. So, it can receive signals from MBS and then transmit to the weak user, that is with lower channel coefficient (i.e., far from MBS), at the same frequency band simultaneously. Denote by N = {$N_{1}$, $N_{2}$, …, $N_{K}$} the set of NOMA groups. There are *V* D2D pairs denoted by D = {$d^{1}$, $d^{2}$, …, $d^{V}$} with $D_{t}$ = {$d_{t}^{1}$, $d_{t}^{2}$, …, $d_{t}^{V}$}, and $D_{r}$ = {$d_{r}^{1}$, $d_{r}^{2}$, …, $d_{r}^{V}$} representing *V* D2D transmitters and *V* D2D receivers, respectively. Only *q* D2D pairs share the sub-channel *k* with each NOMA group $N_{k}$, while $q_{max}$≥q is the maximum number of D2D pairs that a NOMA group can accept to share sub-channel with.

### 3.2. Channel Model

For applying NOMA protocol, at the MBS side, SC is employed to support multiple users to share the same sub-channel, by allocating different power levels for the users. Thus, on the other hand, at CUs side, SIC is employed to decode the superposed signals.

The signal transmitted from MBS side to NOMA group $N_{k}$ on sub-channel *k*∈K is given by:(1)$$x_{k}\;=\;\sum_{i=1}^{m}\sqrt{p_{i,k}}\;x_{i,k},$$
where $x_{i,k}$ is the intended message for the *i*th user in the *k*th NOMA group. $p_{i,k}$ = $\alpha_{i,k}$$P_{k}$ is the allocated power to the *i*th user ${\mathrm{CU}}_{i,k}$, where $P_{k}$ is the maximum transmission power budget of the MBS on a NOMA group $N_{k}$, and $\alpha_{i,k}$∈ (0,1) is the power allocation factor, such that $\sum_{i=1}^{m}\alpha_{i,k}P_{k}\;\leq \;P_{k},\;\forall k\;\in K$.

Without loss of generality, in practice, *m* is set to 2. This is due to the fact that user pairing reduces receiver complexity, reduces SIC error propagation and avoids the extra system coordination overhead that results from large *m* [9]. Therefore, in our system, CUs are paired based on nearest near user and nearest far user (NNNF) scheme as in Reference [10]. This means that, in each NOMA group, the pair consists of MBS-far user ${\mathrm{CU}}_{2,k}$ and MBS-near user ${\mathrm{CU}}_{1,k}$, thus exploiting channel gain difference between ${\mathrm{CU}}_{1,k}$ and ${\mathrm{CU}}_{2,k}$ for successful SIC in each pair. Figure 3 illustrates our system model.

For applying NOMA principle, the strong user ${\mathrm{CU}}_{1,k}$ is allocated less power than the weak user ${\mathrm{CU}}_{2,k}$; thus, the upper limit of $\alpha_{1,k}$ is 0.5. The superposed signal at MBS for NOMA group $N_{k}$ can be written as:(2)$$x_{k}\;=\;\sqrt{p_{1,k}}\;x_{1,k}\;+\;\sqrt{p_{2,k}}\;x_{2,k}.$$

In order to compare the performance of HD mode with FD mode, the model equations are divided into two cases, as described as follows:**Half-Duplex (HD) Mode**:

The HD mode is applied to the strong user (*k* is omitted hereinafter for simplicity, so ${\mathrm{CU}}_{1}$ and ${\mathrm{CU}}_{2}$ refer to the strong user and weak user, respectively, at any sub-channel *k*.) ${\mathrm{CU}}_{1}$, where time division duplexing (TDD) is used for two phases that correspond to odd and even time slots, respectively. During odd time slots, ${\mathrm{CU}}_{1}$ is receiving from MBS, and, during even time slots, ${\mathrm{CU}}_{1}$ is transmitting to ${\mathrm{CU}}_{2}$.

Specifically, in the first phase, corresponding to odd time slots, ${\mathrm{CU}}_{1}$ receives the superposed signal from MBS, and the interference signals of all $d_{t}^{v}$∈$D_{t}$ sharing the same sub-channel *k*, where v∈{1,2,…V} denotes the index of the D2D pair. Therefore, the received signal at ${\mathrm{CU}}_{1}$ is: (The equations with “HD” or “FD” label indicate whether the HD or FD mode is applied, and in the absence of any of these two labels, both modes can be applied.)
(3)$$y_{1,k}^{HD}\;=\;h_{1,k}\left(\sqrt{p_{1,k}}\;x_{1,k}+\sqrt{p_{2,k}}\;x_{2,k}\;\;\right)+\;n\;+\sum_{d\in D_{t}}\;\eta_{d}^{k}\sqrt{P_{d}}\;h_{d,1}\;x_{d},$$
where $x_{d}$ and $P_{d}$ are the transmit signal of D2D transmitter *d* ∈ $D_{t}$ and its transmit power, respectively. *n* ∼ CN(0, *σ*^2^) is the additive white Gaussian noise (AWGN) on sub-channel *k*, and $\eta_{d}^{k}$ is the binary sub-channel allocation coefficient which, if equal to 1, indicates that the D2D transmitter *d* is sharing the same sub-channel of the *k*th NOMA group, and, if equal to 0, it is not. The channel coefficients of channels MBS-${\mathrm{CU}}_{1}$ and ${\text{d-CU}}_{1}$ are denoted by $h_{1,k}$ and $h_{d,1}$, respectively.

In the second phase, corresponding to even time slots, ${\mathrm{CU}}_{1}$ detects ${\mathrm{CU}}_{2}$’s data $x_{2}$ and subtracts it from the received signal via SIC, and then decodes its own data $x_{1}$. So, the received signal-to-interference-and-noise ratio SINR that ${\mathrm{CU}}_{1}$ decodes $x_{2}$ is:(4)$$\gamma_{1,2,k}^{HD}\;=\;\frac{|h_{1,k}{|}^{2}\;p_{2,k}}{|h_{1,k}{|}^{2}\;p_{1,k}\;+\sum_{d\in D_{t}}\;\eta_{d}^{k}\;{\left|h_{d,1}\right|}^{2}P_{d}\;+\;\sigma^{2}},$$
and the received SINR that ${\mathrm{CU}}_{1}$ detects its own data $x_{1}$ is given by:(5)$$\gamma_{1,1,k}^{HD}\;=\;\frac{|h_{1,k}{|}^{2}\;p_{1,k}}{\sum_{d\in D_{t}}\;\eta_{d}^{k}\;{\left|h_{d,1}\right|}^{2}P_{d}\;+\;\sigma^{2}}.$$

In this phase, the strong user ${\mathrm{CU}}_{1}$ decodes and forwards $x_{2}$ to ${\mathrm{CU}}_{2}$ as a way to assist it. Thus, the signal received by ${\mathrm{CU}}_{2}$ includes base station downlink signal, D2D co-channel interference signals, and signal forwarded by ${\mathrm{CU}}_{1}$, and is given by:(6)$$y_{2,k}\;=\;h_{2,k}\left(\sqrt{p_{1,k}}\;x_{1,k}+\sqrt{p_{2,k}}\;x_{2,k}\right)\;+\;n\;+\sum_{d\in D_{t}}\;\eta_{d}^{k}\sqrt{P_{d}}\;h_{d,2}\;x_{d}\;+\;h_{3,k}\sqrt{P_{3,k}}x_{3,k},$$
where $h_{2,k}$, $h_{3,k}$, and $h_{d,2}$ denote the channel gain coefficients of the channels MBS-${\mathrm{CU}}_{2}$, ${\mathrm{CU}}_{1}$-${\mathrm{CU}}_{2}$, and d-${\mathrm{CU}}_{2}$, respectively, $x_{3,k}$ = ${\hat{x}}_{2,k}$ is the decoded message of ${\mathrm{CU}}_{2}$, and $P_{3,k}$ is the transmit power of ${\mathrm{CU}}_{1}$. Assume that the two signals from MBS and ${\mathrm{CU}}_{1}$ are fully resolvable at ${\mathrm{CU}}_{2}$ and can be appropriately co-phased and approximately merged by maximal ratio combining (MRC) [9]. Therefore, the received SINR at ${\mathrm{CU}}_{2}$ to detect data from MBS is written as:(7)$$\gamma_{2,2,k}\;=\;\frac{|h_{2,k}{|}^{2}\;p_{2,k}}{|h_{2,k}{|}^{2}\;p_{1,k}\;+\sum_{d\in D_{t}}\;\eta_{d}^{k}\;{\left|h_{d,2}\right|}^{2}P_{d}\;+\;\sigma^{2}}.$$

The received SINR at ${\mathrm{CU}}_{2}$ to detect data forwarded from ${\mathrm{CU}}_{1}$ is given as:(8)$$\gamma_{2,1,k}\;=\;\frac{|h_{3,k}{|}^{2}\;P_{3,k}}{\sum_{d\in D_{t}}\;\eta_{d}^{k}\;{\left|h_{d,2}\right|}^{2}P_{d}\;+\sigma^{2}}.$$

So, the total SINR at ${\mathrm{CU}}_{2}$ is:(9)$$\gamma_{k,MRC}\;=\;\gamma_{2,1,k}\;+\;\gamma_{2,2,k}.$$

Similarly, the D2D receiver $d_{r}$ receives its intended message from the corresponding D2D transmitter $d_{t}$∈$D_{t}$; and the interference signals from: the base station serving CUs on the same sub-channel *k* that it is assigned to, other D2D transmitters $d_{t}^{\prime }$∈$D_{t}$ on same sub-channel *k*, and the strong user at the even time slot, during which ${\mathrm{CU}}_{1}$-${\mathrm{CU}}_{2}$ transmission occurs (during second phase). So, the received signal at $d_{r}$ is written at the odd and even time slots, respectively, as:(10)$$y_{d_{r},o}^{HD}\;=\;\sqrt{P_{d_{t}}}\;h_{d_{t},d_{r}}x_{d_{t}}\;+\;h_{B,d_{r}}\sqrt{P_{k}}x_{k}+\;n\;+\sum_{d_{t}^{\prime }\in D_{t}\setminus \left\{d_{t}\right\}}\;\eta_{d_{t}^{\prime }}^{k}\sqrt{P_{d_{t}^{\prime }}}\;h_{d_{t}^{\prime },d_{r}}\;x_{d_{t}^{\prime }},$$
and
(11)$$\begin{array}{c} y_{d_{r},e}^{HD}\;=\;\sqrt{P_{d_{t}}}\;h_{d_{t},d_{r}}x_{d_{t}}\;+\;h_{B,d_{r}}\sqrt{P_{k}}x_{k}+\;n\;+\\ \sum_{d_{t}^{\prime }\in D_{t}\setminus \left\{d_{t}\right\}}\;\eta_{d_{t}^{\prime }}^{k}\sqrt{P_{d_{t}^{\prime }}}\;h_{d_{t}^{\prime },d_{r}}\;x_{d_{t}^{\prime }}\;+\;h_{1,k,d}\sqrt{P_{3,k}}\;x_{3,k}, \end{array}$$
where $h_{B,d_{r}}$, $h_{d_{t}^{*},d_{r}}$, and $h_{1,k,d}$ are channel gain coefficients of channels MBS-$d_{r}$, $d_{r}$-$d_{t}^{*}$, and ${\mathrm{CU}}_{1}$-$d_{r}$, respectively; $d_{t}^{*}$ can be $d_{t}$ or $d_{t}^{\prime }$. Therefore, based on (Equation 10) and (Equation 11), the received SINR at $d_{r}$ can be written as (12) and (13), respectively.
(12)$$\gamma_{d_{r},o}^{HD}\;=\;\frac{|h_{d_{t},d_{r}}{|}^{2}\;P_{d_{t}}}{|h_{B,d_{r}}{|}^{2}P_{k}+\sum_{d_{t}^{\prime }\in D_{t}\setminus \left\{d_{t}\right\}}\;\eta_{d_{t}^{\prime }}^{k}\;{\left|h_{d_{t}^{\prime },d_{r}}\right|}^{2}P_{d_{t}^{\prime }}\;+\;\sigma^{2}},$$
(13)$$\gamma_{d_{r},e}^{HD}\;=\;\frac{|h_{d_{t},d_{r}}{|}^{2}\;P_{d_{t}}}{|h_{B,d_{r}}{|}^{2}P_{k}+\sum_{d_{t}^{\prime }\in D_{t}\setminus \left\{d_{t}\right\}}\eta_{d_{t}^{\prime }}^{k}|h_{d_{t}^{\prime },d_{r}}{|}^{2}P_{d_{t}^{\prime }}+{\left|h_{1,k,d}\right|}^{2}P_{3,k}+\sigma^{2}}.$$


**Full-Duplex (FD) Mode:**


Here, the strong user ${\mathrm{CU}}_{1}$ operates using the FD mode, where the MBS-${\mathrm{CU}}_{1}$ and ${\mathrm{CU}}_{1}$-${\mathrm{CU}}_{2}$ channels are non-orthogonal, so it will transmit to ${\mathrm{CU}}_{2}$ on the same sub-channel that it receives on, from MBS simultaneously. So, new interference terms are introduced including the self interference that loops back on ${\mathrm{CU}}_{1}$. So, a change in the channel model equations is introduced as below:

The received signal at ${\mathrm{CU}}_{1}$ will be as follows:(14)$$\begin{array}{c} y_{1,k}^{FD}\;=\;h_{1,k}\left(\sqrt{p_{1,k}}\;x_{1,k}+\sqrt{p_{2,k}}\;x_{2,k}\right)\;\;+\;n\;+\\ \sqrt{\rho }h_{1,1}\sqrt{P_{3,k}}\;x_{3,k}+\sum_{d\in D_{t}}\;\eta_{d}^{k}\sqrt{P_{d}}\;h_{d,1}\;x_{d}, \end{array}$$
where $\sqrt{\rho }h_{1,1}$ indicates the remaining SI level, such that $h_{1,1}$ is the SI channel coefficient, and ρ∈ (0,1] determines the SI level after cancellation because practically, perfect SI cancellation does not exist [26], so, when it is equal to 1, this means that there is no SI cancellation. So, the received SINR that ${\mathrm{CU}}_{1}$ decodes $x_{2}$ is:(15)$$\gamma_{1,2,k}^{FD}\;=\;\frac{|h_{1,k}{|}^{2}\;p_{2,k}}{|h_{1,k}{|}^{2}\;p_{1,k}\;+\sum_{d\in D_{t}}\;\eta_{d}^{k}\;|h_{d,1}{|}^{2}P_{d}\;+\;\rho {\left|h_{1,1}\right|}^{2}P_{3,k}+\;\sigma^{2}},$$
and the received SINR that ${\mathrm{CU}}_{1}$ detects its own data $x_{1}$ is given by
(16)$$\gamma_{1,1,k}^{FD}\;=\;\frac{|h_{1,k}{|}^{2}\;p_{1,k}}{\sum_{d\in D_{t}}\;\eta_{d}^{k}\;|h_{d,1}{|}^{2}P_{d}\;+\;\rho {\left|h_{1,1}\right|}^{2}P_{3,k}+\;\sigma^{2}}.$$

Note that, in FD mode, $d_{r}$ will receive interference from ${\mathrm{CU}}_{1}$ at all times, since ${\mathrm{CU}}_{1}$ is transmitting continually to ${\mathrm{CU}}_{2}$; so, the received signal at D2D receiver $d_{r}$ is written as:(17)$$\begin{array}{c} y_{d_{r}}^{FD}\;=\;\sqrt{P_{d_{t}}}\;h_{d_{t},d_{r}}x_{d_{t}}\;+\;h_{B,d_{r}}\sqrt{P_{k}}x_{k}+\;n\;+\\ \sum_{d_{t}^{\prime }\in D_{t}\setminus \left\{d_{t}\right\}}\;\eta_{d_{t}^{\prime }}^{k}\sqrt{P_{d_{t}^{\prime }}}\;h_{d_{t}^{\prime },d_{r}}\;x_{d_{t}^{\prime }}\;+\;h_{1,k,d}\sqrt{P_{3,k}}\;x_{3,k}. \end{array}$$

Therefore, based on (Equation 17), the received SINR at $d_{r}$ is: (18)$$\gamma_{d_{r}}^{FD}\;=\;\frac{|h_{d_{t},d_{r}}{|}^{2}\;P_{d_{t}}}{|h_{B,d_{r}}{|}^{2}P_{k}+\sum_{d_{t}^{\prime }\in D_{t}\setminus \left\{d_{t}\right\}}\eta_{d_{t}^{\prime }}^{k}|h_{d_{t}^{\prime },d_{r}}{|}^{2}P_{d_{t}^{\prime }}+{\left|h_{1,k,d}\right|}^{2}P_{3,k}+\sigma^{2}}.$$

## 4. Problem Formulation

In this paper, we aim at realizing the spectral efficiency benefits of the proposed interplay, extending coverage, and improving the performance of the weak user. Consequently, this will affect positively the number of assigned D2D pairs. Our objective is to maximize the overall system throughput, while coping with challenging interference to meet the QoS requirements of D2D pairs and CUs, power transmission budget of MBS and SIC constraints. The sum throughput depends strongly on channel allocations; hence, an efficient resource allocation mechanism is necessary. In this section, the optimization problem of sub-channels assignment is formulated.

According to the Shannon Theory, and based on previous SINR equations in Section 3.2, the maximum achievable throughput of ${\mathrm{CU}}_{1}$, ${\mathrm{CU}}_{2}$, and the D2D receiver $d_{r}$, respectively, are given as:(19)$$R_{1,k}\;=\mu \;\log_{2}\;\left(\;1\;+\;\gamma_{1,1,k}\;\right),$$
(20)$$R_{2,k}\;=\mu \;\min\;\left\{\log_{2}\left(1+\gamma_{1,2,k}\right)\;,\;\log_{2}\left(1+\gamma_{k,MRC}\right)\right\},$$
and
(21)$$R_{d_{r}}=\left\{\begin{array}{cc} R_{d_{r}}^{HD} & =\frac{1}{2}\log_{2}\;\left(\;1+\gamma_{d_{r},o}^{HD}\;\right)\;+\;\frac{1}{2}\log_{2}\;\left(\;1+\gamma_{d_{r},e}^{HD}\;\right),\;\mathrm{HD}\;\mathrm{mode}\;\\ R_{d_{r}}^{FD} & =\log_{2}\;\left(\;1+\gamma_{d_{r}}^{FD}\;\right),\;\mathrm{FD}\;\mathrm{mode} \end{array}\right.,$$
where $\mu \in \{\frac{1}{2},1\},$ it is equal to $\frac{1}{2}$ in HD mode and 1 in FD mode. Note that $\gamma_{1,1,k}$ and $\gamma_{1,2,k}$ will be calculated based on Equations (Equation 4) and (Equation 5), respectively, in HD mode and based on Equations (Equation 15) and (Equation 16), respectively, in FD mode.

So, the total network throughput is defined as the sum of cellular users’ and D2D receivers’ data rates, it is given by:(22)$$R_{sum}\;=\;\sum_{k=1}^{K}\left(R_{1,k}\;+\;R_{2,k}\;+\;\sum_{d_{r}\in D_{r}}\eta_{d_{r}}^{k}R_{d_{r}}\right).$$

### Optimization Problem Formulation

Based on the previous equations, the optimization problem is formulated as below:

(23)$$\begin{array}{cccc} \mathrm{P1:} & \max_{\eta ,\alpha_{k}} & R_{sum}\left(\;\eta \;,\;\alpha_{k}\;\right) & \\ \; & s.t. & C_{1}:\;\gamma_{1,1,k}\geq \gamma_{1}^{thresh}, & \;\forall k,\\ \; & \; & C_{2}:\;min\left(\gamma_{k,MRC},\gamma_{1,2,k}\right)\geq \gamma_{2}^{thresh}, & \;\forall k,\\ \; & \; & C_{3}:\;\gamma_{1,2,k}\geq \gamma_{2,2,k}, & \;\forall k,\\ \; & \; & C_{4}:\;\gamma_{d_{r}}\geq \gamma_{d_{r}}^{thresh}, & \;\forall d\in D,\\ \; & \; & C_{5}:\;\eta_{d}^{k}\in \left\{0,1\right\}, & \;\forall d,\forall k,\\ \; & \; & C_{6}:\;\sum_{k}\eta_{d}^{k}=1, & \;\forall d,\\ \; & \; & C_{7}:\;\alpha_{i,k}\geq 0, & \;\forall k,\;i=1,2,\\ \; & \; & C_{8}:\;\sum_{i=1}^{2}\alpha_{i,k}\leq 1, & \;\forall k, \end{array}$$
where $\gamma_{1}^{thresh}$ and $\gamma_{2}^{thresh}$ are the received SINR thresholds of the strong user and weak user in each NOMA group, respectively, and they are equal to $2^{R^{thresh}}-1$ in FD mode and $2^{2R^{thresh}}-1$ in HD mode, where $R^{thresh}$ is the rate threshold of the cellular users. Similarly, $\gamma_{d_{r}}^{thresh}$ = $2^{R^{threshd}}-1$ and $\gamma_{d_{r}}$ = $2^{R_{d_{r}}}-1$ are the received SINR threshold of $d_{r}$ and received SINR at $d_{r}$, respectively, where $R^{threshd}$ is the rate threshold of the D2D pairs.

Now, in order to maximize the system sum rate, constraints of the problem in (23) must be considered. Constraints **C**_1_, **C**_2_ and **C**_4_ denote QoS requirements of NOMA group users and D2D receivers. Hence, by considering them, interference levels are limited by keeping their received SINR greater than the corresponding SINR threshold. Constraint **C**_3_ ensures the SIC success in each NOMA group. Constraint **C**_5_ shows that $\eta_{d}^{k}$ is a binary variable that takes two values when D2D pair is sharing sub-channel *k* (1), and when it it is not (0). Constraint **C**_6_ ensures that each D2D pair can operate on only one sub-channel. Finally, constraints **C**_7_ and **C**_8_ are related to power allocation of NOMA group users, so the first ensures that the power allocation factor is positive, and the last ensures that the total power does not exceed the maximum transmission power budget.

The formulated problem **P1** in (23) deals with non-convex objective function [27], which introduces intractability in solving. Hence, it is computationally complicated to reach the optimal solution of this problem.

A swapping-based matching theory algorithm is implemented for allocating sub-channels in the proposed system, similar to Reference [18] and to the extended work of Reference [4] (Reference [28]). Knowing that matching theory depends on arrangement of two sets of players based on their preferences (utilities), utilities calculation will be based on the achievable rates in this FD NOMA-based cellular cooperative scenario with underlaying D2D. Implementation of Pareto-based matching theory aims to increase total sumRate while finding the best matching, but, at the same time, ensuring that any individual sumRates of CUs and D2D pairs is not decreasing. In Section 5, the sub-channel allocation algorithm between NOMA groups and D2D pairs will be explained based on a fixed power allocation that guarantees SIC success requirements.

## 5. Sub-Channels Allocation

At this stage, we suppose that the power allocated to the CUs in NOMA groups is fixed so that $\alpha_{1,k}<0.5$. The problem of sub-channels assignment can be formulated as:
(24)$$\begin{array}{ccc} P2: & \max_{\eta } & R_{sum}\left(\;\eta \;\;\right)\\ \; & s.t. & C_{1}-C_{6}. \end{array}$$


Due to the fact that there are inter-dependencies between users in the formulated problem **P2** and that finding solution is difficult in a practical amount of time, especially for a large number of users and sub-channels, the sub-channel allocation is formulated using matching theory. It is formulated as a *many-to-one two-sided matching game with externalities* between D2D pairs and sub-channels (or NOMA groups because each NOMA group is assigned to a sub-channel). So, *each NOMA group can be matched to many D2D pairs over one sub-channel, and each D2D pair can be matched to one NOMA group over one sub-channel*, and *externalities* are due to the interference relations that are presented in channel model equations.

### 5.1. Matching Theory Concepts

Now, matching theory concepts are given. The matching will be defined as follows:

**Definition** **1.**
*A many-to-one matching ψ is defined over the set D∪N, and it is characterized by:*
*1*. 
*$|\psi (d^{v})|$ = 1, ∀$d^{v}$∈D and $\psi (d^{v})\in N$,*
*2*. *$|\psi (N_{k})|$$\leq q_{max}$,* ∀$N_{k}$∈N
*and $\psi (N_{k})\subset D$,*
*3*. 
*$\psi (d^{v})$ = $N_{k}$ if $d^{v}$$\in \psi (N_{k})$,*
*4*. 
*$\psi (N_{k})$ = $S_{d}\subset D$ = {$d^{v}$∈D : $\psi (d^{v})$ = $N_{k}$}.*



Based on **Definition 1**, it is clear from (1.) and (2.) that each D2D pair can be matched to only one NOMA group, and each NOMA group can be matched to a maximum of $q_{max}$ D2D pairs.

In this *two-sided matching game*, D2D pairs and NOMA groups act as two sets of players. Each player has a list of preferences on the players of the other set. Preferences (i.e., indexes of satisfaction) represent how each player is satisfied by each player of the other set, in case they are matched. So, the matching is given the tuple ($D,N,P_{D},P_{N}$), where $P_{D}$ and $P_{N}$ represent the preference lists of D2D pairs and NOMA groups, respectively.

The preferences can be quantified by QoS values [8]. So, a *preference function* calculates, for each player, the QoS value achieved when it is matched with players from the other set. Consequently, each player ranks players of the other set based on these calculated QoS values. For this aim, two utility functions $U_{d^{v}}\left(N_{k}\right)$ and $U_{N_{k}}\left(S_{d}\right)$ for D2D pairs and NOMA groups, respectively, are defined. The following utility functions determine the preferences of the players.
(25)$$U_{d^{v}}\left(N_{k}\right)\;=\;R_{d_{r}^{v}}\;$$
(26)$$U_{N_{k}}\left(S_{d}\right)\;=\;R_{1,k}\;+\;R_{2,k}+\;\sum_{d_{r}\in S_{d}}R_{d_{r}}.$$

Equation (Equation 25) represents for D2D pair $d^{v}$, the utility on NOMA group $N_{k}$, and it is equal to acheivable data rate of $d^{v}$, when it is matched to sub-channel k (shared with $N_{k}$). On the other side, (Equation 26) represents for NOMA group $N_{k}$, the utility on a set of D2D pairs $S_{d}$ ($|S_{d}|\leq q_{max}$) sharing sub-channel *k*, and it is equal to the total sumRate on sub-channel *k*.

The goal is to find the best two-sided stable matching. For reaching this stable matching, swappings are permitted, so D2D pairs can swap their sub-channels. So, first, to better define swapping, the following process is given:Consider a matching ψ and take a pair of D2D pairs ($d^{v}$,$d^{v^{\prime }}$) in this matching, let $d^{v}$ be matched to $N_{k}$ (so $\psi \left(d^{v}\right)=N_{k}$), and $d^{v^{\prime }}$ matched to $N_{k^{\prime }}$ (so $\psi \left(d^{v^{\prime }}\right)=N_{k^{\prime }}$).After swapping, the two D2D pairs switch their matchings, so, $\psi \left(d^{v}\right)=N_{k^{\prime }}$ and $\psi \left(d^{v^{\prime }}\right)=N_{k}$.All other D2D pairs and NOMA groups (sub-channels) in ψ are not affected by swapping of $d^{v}$ and $d^{v^{\prime }}$, and their matchings are kept the same.Thus, a new matching $\psi_{v}^{v^{\prime }}$ will be the same as the old matching ψ, but with $d^{v}$ and $d^{v^{\prime }}$ swapping their matchings.

A matching is two-sided stable if no *swap-blocking pair* exists. Based on concept of *Pareto improvement*, the *swap-blocking pair* is defined as below with two conditions:

**Definition** **2.***$\left(d^{v},d^{v^{\prime }}\right)$ is a swap-blocking pair if and only if**1*. ∀i∈ {$d^{v}$, $d^{v^{\prime }}$, $N_{k}$, $N_{k^{\prime }}$}, $U_{i}\left(\psi_{v}^{v^{\prime }}\right)$ ≥ $U_{i}\left(\psi \right)$ and*2*. ∃i∈ {$d^{v}$, $d^{v^{\prime }}$, $N_{k}$, $N_{k^{\prime }}$}, $U_{i}\left(\psi_{v}^{v^{\prime }}\right)$ > $U_{i}\left(\psi \right)$,
where $N_{k}=\psi \left(d^{v}\right)$ and $N_{k^{\prime }}=\psi \left(d^{v^{\prime }}\right)$; $\psi_{v}^{v^{\prime }}$ is the new matching after swapping between $d^{v}$ and $d^{v^{\prime }}$; and $U_{i}\left(\psi \right)$ and $U_{i}\left(\psi_{v}^{v^{\prime }}\right)$ are the utilities of element *i* in the case of matching ψ and $\psi_{v}^{v^{\prime }}$, respectively.

Based on **Definition 2**, for the swapping process to be done, *Pareto improvement* conditions must be satisfied. So, utilities of all players involved in swapping must not be reduced (condition 1), and utility of at least one involved player must be increased (condition 2). This guarantees that, after any swapping operation, the system sum rate will increase, until reaching Pareto efficiency, i.e., no existence of swap-blocking pairs.

### 5.2. Sub-Channels Allocation Algorithm

A two-sided many-to-one matching algorithm based on Pareto improvement is implemented and presented in Algorithm 1.

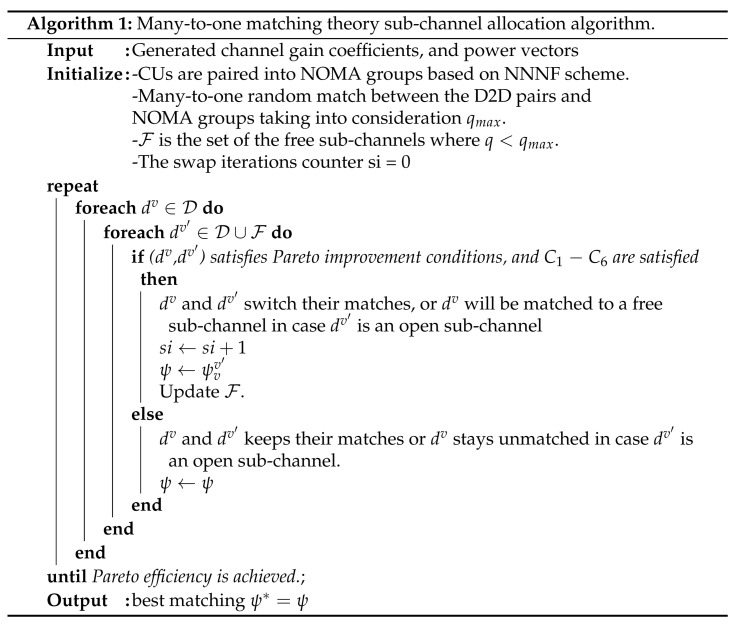


First, NNNF scheme is used to pair CUs into NOMA groups, then each NOMA group is assigned to a sub-channel. Swappings will be based first on initial random matching between D2D pairs and groups of the cellular users. This random matching can include unassigned D2D pairs or free sub-channels (number of assigned D2D pairs on free sub-channels is less than the maximum allowed, i.e., $q_{max}$). Then, each D2D pair searches for a potential D2D pair that can form a swap-blocking pair with (satisfy Pareto improvement conditions that ensure increasing the total sumRate at each swapping). If the conditions are satisfied, the two pairs swap their matching, and all other matchings are kept the same. The swapping between the two pairs can lead to the case of having one of them unassigned to a sub-channel, thus allowing for initially unassigned D2D pairs to be assigned again.

The finite number of CUs and D2D pairs results in a finite number of swap-blocking pairs and, thus, finite number of swapping iterations, leading to achievement of Pareto efficiency and reaching the best matching.

## 6. Simulation Results and Analysis

In this section, numerical results are presented using MATLAB simulations to evaluate the overall system performance. The CUs and D2D pairs are randomly distributed within the MBS coverage. Figure 4 shows a snapshot of a uniform cellular distribution with MBS at the center of the cell and radius of 100 m. The simulation results are depicted using 1000 random cellular distributions, and average values are plotted. Strong users are randomly located within a disk of radius 50 m, and weak users were randomly located within a ring of radii 85 and 100 m. This assumption is due to the fact that the strong users are considered to be near the MBS, and the weak users are far from it. The D2D pairs were randomly located in the whole cell. In each NOMA group, CUs were paired using NNNF scheme [10], and each NOMA group was randomly assigned with a sub-channel *k*. The path loss function used for simulation is $PL\left(distance\right)=distance^{-PLE}$, where PLE denotes the path loss exponent. Table 2 shows the simulation parameters used, unless otherwise specified. In addition, all simulation results are implemented in FD cooperative NOMA (CNOMA), unless otherwise specified, for comparison aim.

Figure 5 shows the variation in number of swapping iterations with number of D2D pairs, while number of NOMA groups (or cellular users) is kept constant. Note that, hereinafter, “number of D2D pairs” indicates to all D2D pairs in the cell, both the assigned and not assigned D2D pairs. As shown, the number of swapping iterations increases as number of D2D pairs increases. This is due to the fact that, with the increase in present number of D2D pairs, possibilities of having more swap-blocking pairs that satisfy Pareto improvement conditions increase.

In order to show the benefits of cooperative NOMA with respect to the conventional one (i.e., non-cooperative), we plot in Figure 6 the variation of the number of assigned D2D pairs as a function of the total number of D2D pairs in the cell for both cooperative and non-cooperative NOMA, considering both one-to-one and many-to-one matching theory for sub-channels allocation.

As previously mentioned, the number of assigned D2D pairs is determined by the number of D2D pairs that can share sub-channels with NOMA groups, and this is by considering QoS requirements and the remaining problem constraints. The results of Figure 6 show that, as the number of D2D pairs increases, the probability of having D2D pairs that increase the total throughput, while respecting the constraints at the same time, increases. Moreover, to compare one-to-one matching theory, that allows only one D2D pair to share the sub-channel with the NOMA group, with many-to-one, it is clear in the figure that the latter outperforms the former. In both, the number of assigned D2D pairs starts increasing in a fast rate, until the number of D2D pairs reaches the *concentration point* (i.e., ∼7 in one-to-one and ∼14 in many-to-one) at which the rate of increase starts to slow down or to be approximately zero. This is due to the fact that the maximum number of assigned D2D pairs in the whole cell is equal to $q_{max}$*K (knowing that K = 7, and $q_{max}$ = 1 for one-to-one and $q_{max}$ = 2 for many-to-one).

Concerning the performance of cooperative and non-cooperative NOMA, it is shown in Figure 6 that, with both many-to-one and one-to-one matching, the cooperative NOMA allows a greater number of assigned D2D pairs than that with non-cooperative NOMA. When D2D pairs share the sub-channel with NOMA groups, the weak user in each NOMA group will be affected with interference negatively more than the strong user. Thus, the performance of the weak user will have the dominant influence on analysis of the number of assigned D2D pairs.

With non-cooperative NOMA, the weak user is not assisted, and consequently its received SINR is not improved, so as the interference level on it increases, its received SINR is easily lowered below the received SINR threshold. Therefore, it will not allow more D2D pairs to share the sub-channel to prevent the additional interference that lowers the SINR below the threshold. On the contrary, with cooperative NOMA, the weak user is assisted and its SINR is improved, so, more D2D pairs will be allowed to be assigned to sub-channels shared by the cellular users. Interestingly, the difference in performance between both is greater at greater number of D2D pairs. This highlights a higher advantage for cooperative over non-cooperative NOMA with a denser D2D tier (i.e., greater number of D2D pairs). Thus, through interpreting this result, the goal of cooperative NOMA for improving performance of the cell-edge users (weak users), thus boosting reception reliability and extending coverage, is proved.

Figure 7 also studies the variation of the number of assigned D2D pairs as a function of the SINR threshold of cellular users. It is clearly observed that higher the SINR threshold of cellular users is, the less number of assigned D2D pairs is. This returns to that, with increasing the QoS requirements of cellular users, the maximum allowed interference level on cellular users decreases, so the probability of having D2D pairs that respect the SINR threshold of the cellular users decreases. Consequently, the number of assigned D2D pairs decreases, and more D2D pairs will be left unassigned to sub-channels, or more sub-channels are needed to let them access the network. This scenario applies also to non-cooperative NOMA, but it is affected more negatively than that our work that is using cooperative NOMA, due to the low received SINR at the weak user.

Moreover, Figure 7 shows the variation of the result while changing the SI cancellation level ρ. We recall that ρ=0 refers to the perfect SI cancellation scenario, while, when it is equal to 1, to the no SI cancellation. The results show that, when ρ increases, the number of assigned D2D pairs decreases. This is due to the fact that the influence of self interference increases; thus, the whole interference exceeds the maximum allowed interference, and the received SINR lowers down the threshold. Consequently, NOMA groups will no longer accept D2D pairs, in order to keep their QoS requirements. Furthermore, note that, at low SINR threshold values of cellular users, the difference in the result between different ρ values is small, while this difference increases with higher SINR thresholds.

Figure 8 and Figure 9 show the variation of the total throughput with the total number of D2D pairs, and with maximum D2D transmission distance, respectively. Each result is applied with three different scenarios, i.e., first is many-to-one matching theory, second is one-to-one matching theory, and third is random many-to-one allocation, where, in the third, up to two D2D pairs are randomly assigned to each sub-channel shared by a NOMA group.

In Figure 8, with the first two scenarios, it is observed that as the number of D2D pairs increases, the total sumRate increases, due to the increase in number of assigned D2D pairs as shown in Figure 6. Concerning the first scenario that is implemented in this work, it is observed that, even when the number of D2D pairs exceeds 14 ($q_{max}$*number of sub-channels), the total sumRate remains increasing due to the increase in probability of having more D2D pairs that can be assigned to the sub-channel and increase the total sumRate. Similarly, this applies to the second scenario (using one-to-one matching theory). It is shown that the first has achieved more gain than the second in terms of total sumRate; it is also clearly seen that, with 9 D2D pairs, approximately, there is no difference in the result between them, this difference starts increasing when the number of D2D pairs increases. The results also show that these two outperforms the random many-to-one allocation. It is added for comparison to prove that one-to-one matching theory allocation with the many-to-one matching theory is worth comparing.

Figure 9 shows that as the distance between D2D transmitter and D2D receiver increases, the total sumRate decreases. This negative relation is due to the additional path loss that the transmitted signal of the D2D transmitter will face with more crossed distance can affect negatively the received SINR of D2D receiver. Consequently, the total sumRate is affected. The same analysis of Figure 8 of comparing the three scenarios applies here, where that implemented in this work outperforms the other two.

Figure 10 shows the variation of total sumRate with the SINR threshold of cellular users and the transmit power of D2D pairs. It is shown that the total sumRate increases as the transmit power of D2D pairs increases; this is because more transmit power at D2D transmitters leads to a greater received SINR at the D2D receivers, thus affecting total sumRate positively. But, after certain increase in this transmit power, the total sumRate starts to decrease, and this can be understood because more D2D transmit power means more interference on the cellular users lowering their received SINR down. Moreover, regarding the influence of variation of cellular users SINR thresholds on the second axis, when it increases, the total sumRate decreases due to decrease in number of assigned D2D pairs as stated in Figure 7 at 20 dBm of transmit power of D2D pairs and this applies to other values of transmit power. As shown in Figure 10, it is worth it to mention that this decrease is not present or not significant at low transmit power of D2D pairs because the interference on cellular users in this case will not be severe enough to pull their received SINR below the thresholds. So, the SINR thresholds are not acting a dominant role at low transmit power of D2D pairs, but it returns its role with higher transmit power of D2D pairs, where the decrease in total sumRate with increase of SINR threshold is clear.

Figure 11 shows the variation of the number of assigned D2D pairs with transmit power of D2D pairs, and it compares the result between full-duplex cooperative NOMA (FD CNOMA) and half-duplex cooperative NOMA (HD CNOMA) at different Cellular users rate thresholds. Considering the same rate threshold of cellular users $R^{thresh}$, it is observed that FD mode achieves higher number of assigned D2D pairs than HD mode. Thus, in HD mode, a greater number of D2D pairs will remain unassigned, or more sub-channels are needed to accommodate them. This return to the fact $\gamma_{1}^{thresh}$ and $\gamma_{2}^{thresh}$ are equal to $2^{R^{thresh}}-1$ and to $2^{2R^{thresh}}-1$ in FD and HD mode, respectively, and taking into consideration the negative relation between cellular users SINR threshold and number of assigned D2D pairs in Figure 7. In addition, keeping the same $R^{thresh}$, it can be seen that the difference in result between FD CNOMA and HD CNOMA increases as the transmit power of D2D pairs increases because the additional interference will affect the HD CNOMA more negatively. This difference between FD CNOMA and HD CNOMA over all values of transmit power of D2D pairs is higher with $R^{thresh}$ = 0.5 than that with $R^{thresh}$ = 0.1, also for the same reason.

The results are not strictly decreasing. It is observed that number of assigned D2D pairs starts increasing at low transmit power, then decreases. This is due to the fact that, at low transmit power of D2D pairs, as it increases, the received SINR at D2D pairs increases. At the same time, this low power does not add severe interference that can compete with SINR threshold of cellular users. This allows more D2D pairs to access the sub-channels shared by NOMA groups, while keeping their QoS requirements, without being affected or affecting CUs by high interference levels. Moreover, when the transmit power exceeds a certain value, D2D pairs start to introduce more interference. Thus, CUs will no longer accept more D2D pairs to be assigned to their sub-channels. Contrary to that, this does not apply in HD CNOMA with $R^{thresh}=0.5$; instead, the result is degraded directly by the interference introduced by D2D pairs.

Figure 12 and Figure 13 compare FD and HD modes in terms of the total sumRate and the number of assigned D2D pairs with the number of D2D pairs.

In Figure 12, at $R^{thresh}$ = 0.1 bits/s, first, in terms of total sumRate, it is shown that HD mode outperforms the FD mode at small number of D2D pairs, and then it starts to decrease sharply, while, in FD mode, sumRate is still increasing with number of D2D pairs so the latter exceeds the former at denser D2D tier (at higher number of D2D pairs). This variation owes to that the HD mode achieves higher total sumRate at small number of assigned D2D pairs, unlike the FD mode.

Similarly, in Figure 13, at $R^{thresh}$ = 0.3 bits/s, the same scenario applies, where, in HD mode, the total sumRate starts to decrease, then increases slowly until the difference in total sumRate between HD and FD mode decreases as the D2D tier becomes denser, while, at the same time, FD mode keeps a higher number of assigned D2D pairs.

Proceeding from the two figures Figure 12 and Figure 13, at higher rate threshold for cellular users, HD CNOMA achieves higher total sumRate than FD CNOMA, where this difference decreases as the D2D tier becomes denser. On the other hand, FD CNOMA achieves higher number of assigned D2D pairs while keeping significant difference with HD CNOMA, while, at lower rate threshold, FD mode starts to achieve a higher total sumRate at denser D2D tier and also keeping higher number of assigned D2D pairs but with lower difference with HD mode.

Figure 14 shows the influence of the self interference channel gain with different self interference cancellation factors. As observed, as the self interference channel gain increases, the total sumRate decreases. This is due to the fact that, when the self interference channel gain increases, the received SINR values at the strong and weak users will be negatively affected and this leads to decrease in total sumRate. In addition, it is shown that, with a greater level of self interference cancellation 1 − ρ, a greater total sumRate will be achieved, and this is due to the fact that the greater number of assigned D2D pairs will be, in case of smaller ρ as shown in Figure 7, leading to a decrease in the total sumRate. The rate of this decrease increases with a higher value of ρ, due to high increase in interference and, thus, a steep decrease in number of assigned D2D pairs.

In general, it is observed that multiple D2D pairs assigned with the same NOMA group on a sub-channel is not necessarily improving the total performance more than having only one D2D pair for each sub-channel. This is shown by the random many-to-one allocation added for comparison with matching-theory based allocations (one-to-one and many-to-one). This is due to the fact that, without methodical interference management, many D2D pairs on one sub-channel can instead deteriorate the performance. This is especially on the level of the weak cellular user, which cooperative NOMA has been observed to improve its performance compared to the non-cooperative NOMA.

Specifically, through results analysis, a trade off lies between QoS requirements of CUs and the total performance especially at higher transmit power of D2D pairs. Moreover, QOS requirements are observed to affect the performance difference between FD and HD modes. Furthermore at last, a denser D2D tier that is more realistic has given advantage for the proposed system and method over others.

## 7. Conclusions

In this work, the application of full-duplex cooperative NOMA system and underlaying D2D communication is investigated. The goal was to maximize the sumRate, extend the coverage by improving the performance of cell-edge (far) users with introducing the cooperative relaying, as well as increase number of assigned D2D pairs. So, resource allocation is applied, while coping with the challenging interference and keeping QoS, as well as SIC requirements. In simulation results, it is shown that implementing sub-channels allocation with many-to-one matching theory allocation outperforms that by the one-to-one matching or random matching, where the gain of the first over the other two is highlighted significantly with a denser D2D tier. Furthermore, the goal of cooperative relaying is proved, and it is compared with the non-cooperative NOMA scenario. In addition, the effect of self interference with FD mode operating at the strong user on the overall performance is investigated. Along with these results, FD and HD modes are compared in terms of total sumRate and number of assigned D2D pairs, where the difference in their performance has been proved to depend on the rate threshold of the cellular users. Moreover, although FD mode is not always outperforming HD in terms of total sumRate, a denser D2D tier gives an advantage for FD over HD mode.

## Figures and Tables

**Figure 1 sensors-21-02768-f001:**
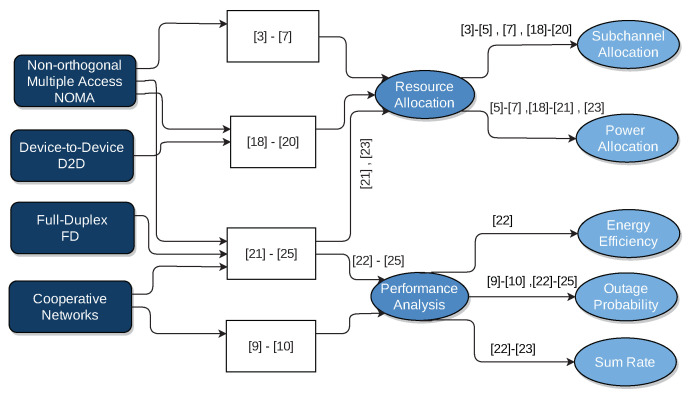
Related works.

**Figure 2 sensors-21-02768-f002:**
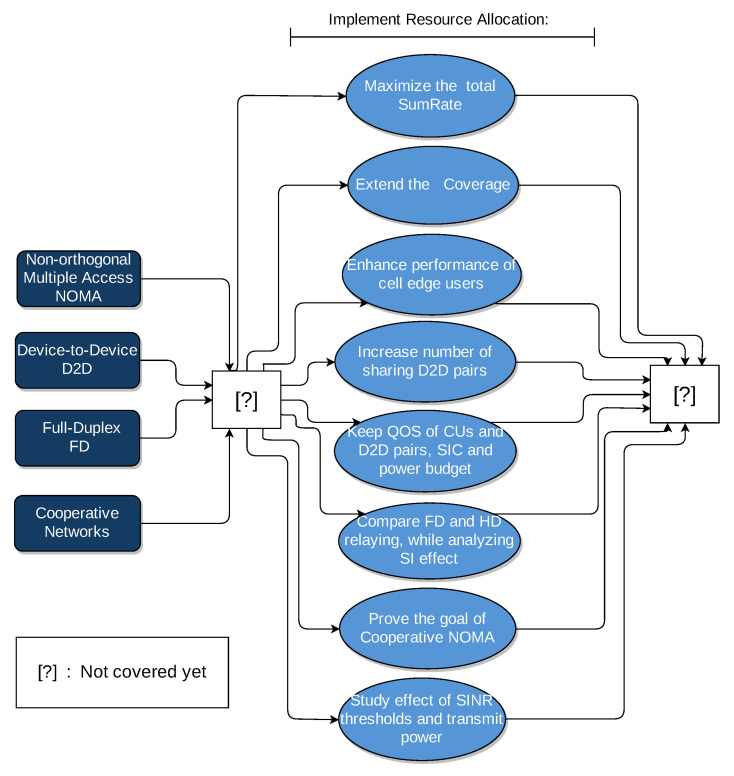
Contribution of this work.

**Figure 3 sensors-21-02768-f003:**
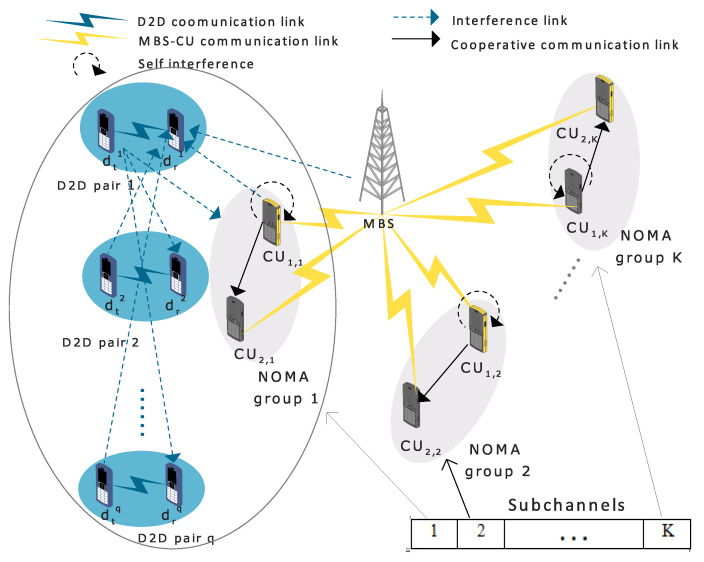
System model for single-cell downlink transmission scenario.

**Figure 4 sensors-21-02768-f004:**
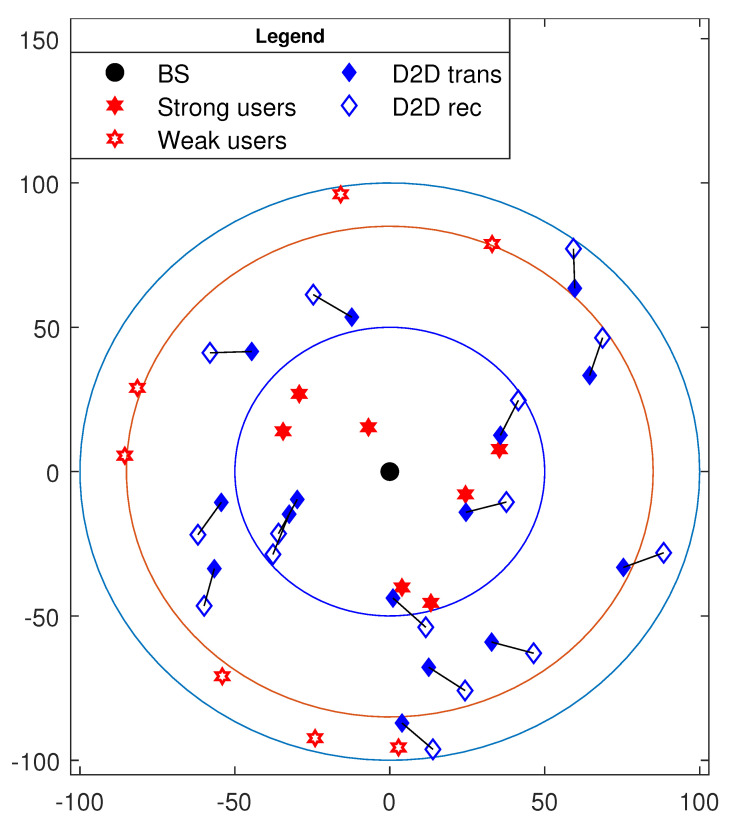
A snapshot of single-cell distribution with 7 non-orthogonal multiple access (NOMA) groups (14 cellular users) and 15 device-to-device (D2D) pairs.

**Figure 5 sensors-21-02768-f005:**
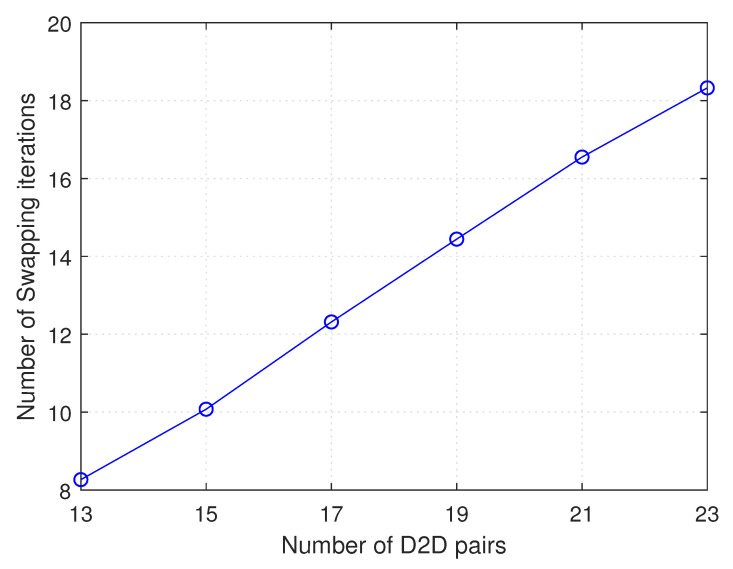
Number of swapping iterations versus number of D2D pairs.

**Figure 6 sensors-21-02768-f006:**
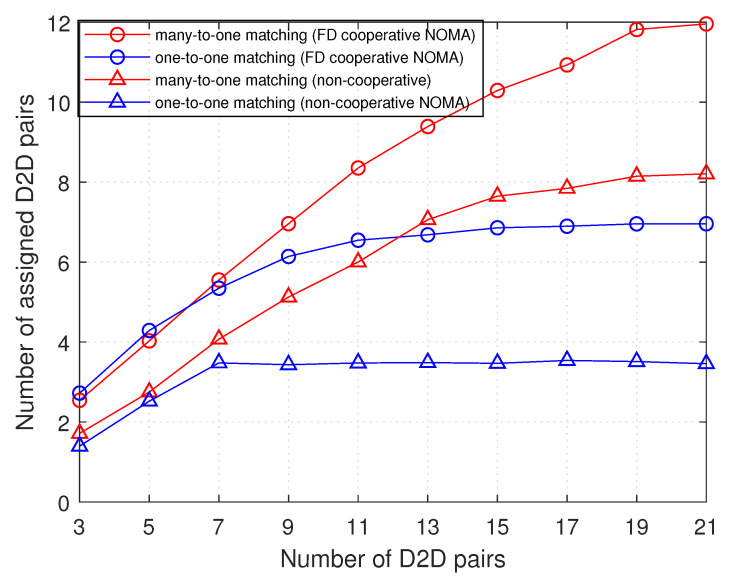
Assigned D2D pairs versus number of D2D pairs.

**Figure 7 sensors-21-02768-f007:**
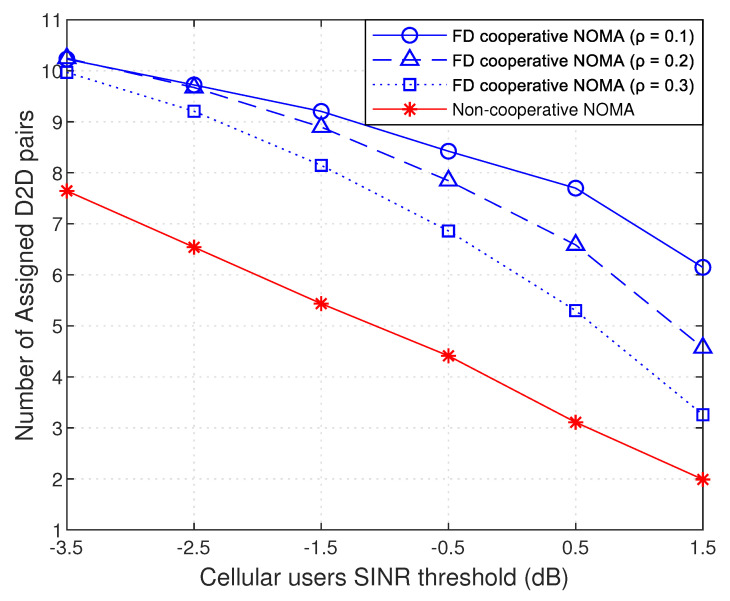
Assigned D2D pairs versus cellular users SINR threshold.

**Figure 8 sensors-21-02768-f008:**
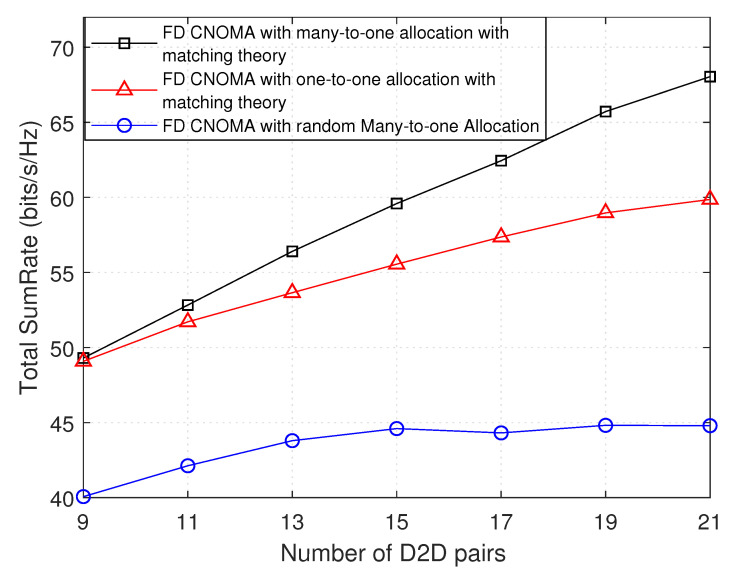
Total sumRate versus number of D2D pairs.

**Figure 9 sensors-21-02768-f009:**
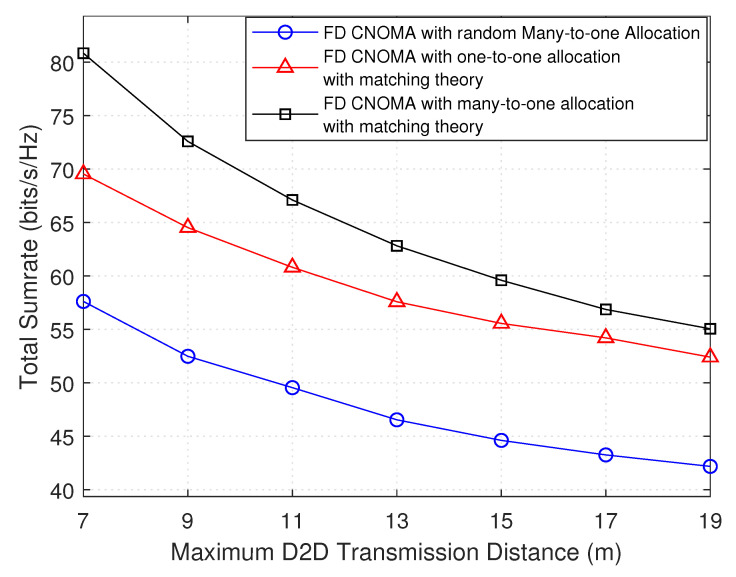
Total sumRate versus maximum D2D transmission distance.

**Figure 10 sensors-21-02768-f010:**
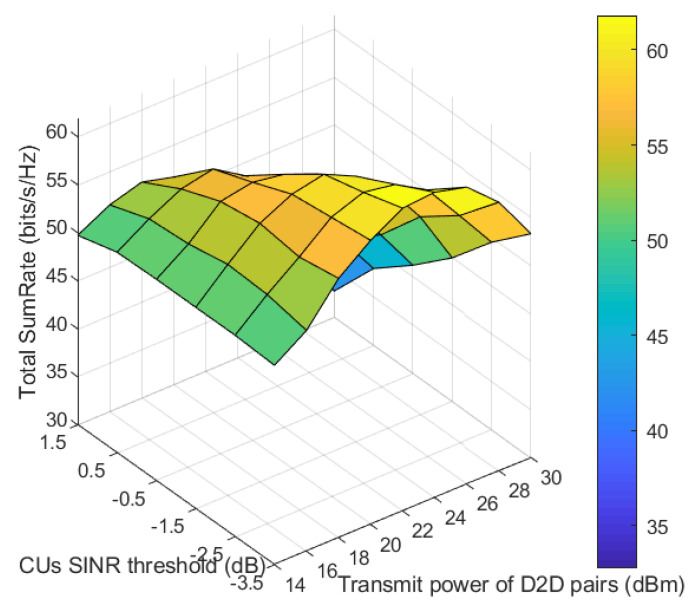
Total sumRate versus D2D transmit power and cellular users (CUs) SINR threshold.

**Figure 11 sensors-21-02768-f011:**
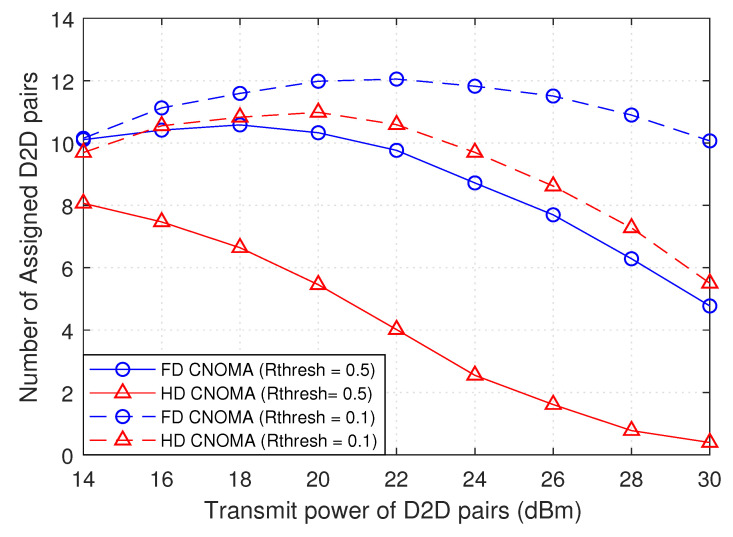
Number of assigned D2D pairs versus transmit power of D2D pairs with full-duplex (FD) and half-duplex (HD) modes at different rate thresholds $R^{thresh}$(bits/s/Hz) of CUs.

**Figure 12 sensors-21-02768-f012:**
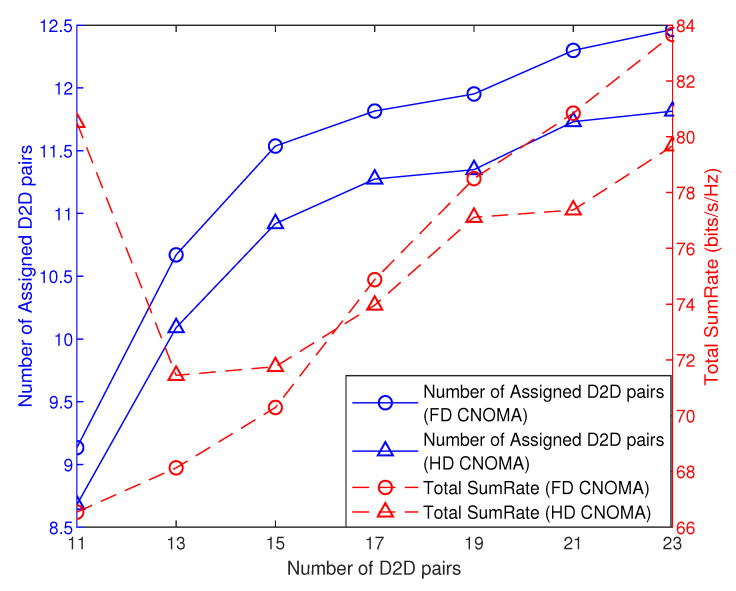
Total sumRate and number of assigned D2D pairs versus number of D2D pairs ($R^{thresh}=0.1$, $|h_{1,1}{|}^{2}=-20$, ρ=0.01).

**Figure 13 sensors-21-02768-f013:**
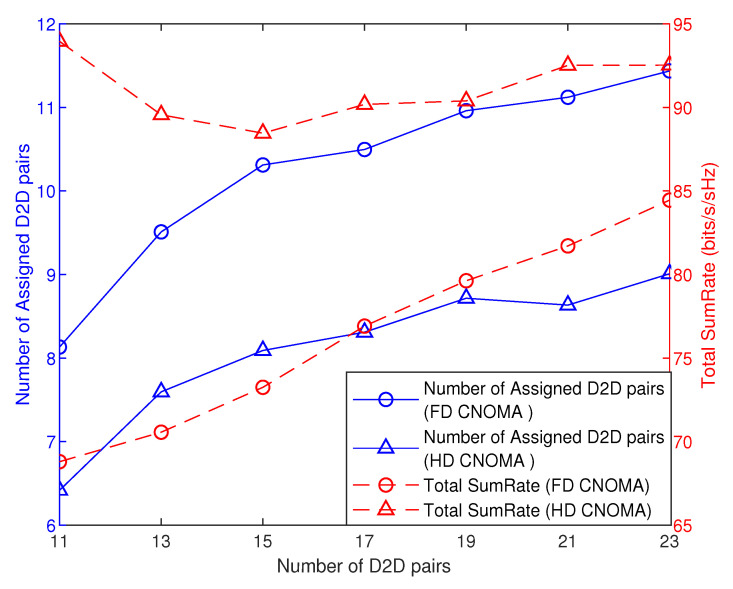
Total sumRate and number of assigned D2D pairs versus number of D2D pairs ($R^{thresh}=0.3$, $|h_{1,1}{|}^{2}=-20$, ρ=0.01).

**Figure 14 sensors-21-02768-f014:**
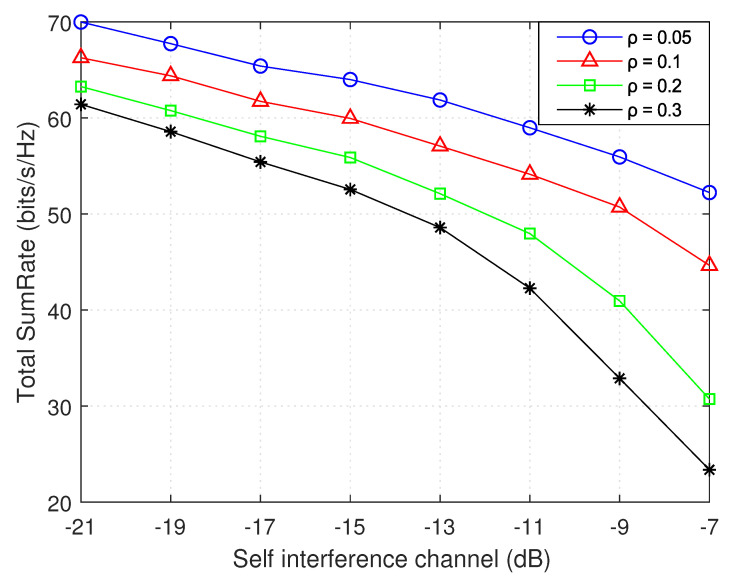
Total sumRate versus self interference level.

**Table 1 sensors-21-02768-t001:** Notation table.

Notations	Definitions
U,C	number and set of cellular users
K, K	number and set of sub-channels
N	set of NOMA groups
V, D	number and set of D2D pairs
$CU_{1,k},CU_{2,k}$	strong and weak user of the $k^{th}$ NOMA group
α	power allocation coefficient
h	Channel gain coefficient (path loss and Rayleigh fading)
P	Transmission power
ρ	Self interference cancellation factor

**Table 2 sensors-21-02768-t002:** Simulation parameters.

Parameters	Values
Cell Radius	100 m
Pathloss Exponent *PLE*	2
Maximum D2D transmission distance $d_{max}$	15 m
Noise power $\sigma^{2}$	−118 dBm
Maximum transmission power of MBS on each sub-channel	30 dBm
Transmission power of the cooperating user	10 dBm
Transmission power of D2D pairs	20 dBm
Rate threshold of Cellular users $R^{thresh}$	0.5 bits/s/Hz
Rate threshold of D2D pairs $R^{threshd}$	0.5 bits/s/Hz
Number of sub-channels K	7
The self interference channel $|h_{1,1}{|}^{2}$	−15 dB
The self interference cancellation factor ρ	0.1
Maximum number of assigned D2D pairs/sub-channel: $q_{max}$	2

## Data Availability

Not applicable.

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
