# Peer review of "Resource Allocation for Downlink Full-Duplex Cooperative NOMA-Based Cellular System with Imperfect SI Cancellation and Underlaying D2D Communications"

_sensors, 2021, doi:10.3390/s21082768_

Round 1
Reviewer 1 Report
In this paper, the authors propose a full-duplex cooperative NOMA-based cellular system to optimize the system sum rate by considering the following constraints: QoS, power budget, and interference cancellation.
The paper is well-written and interesting to read, however I would like to include some comments as follows:
- What is the main reason behind considering the downlink transmission instead of the uplink or both the scenarios?
- Single-cell transmission scenario is quite easy to analyze. Have you ever try to implement your idea on multi-cell transmission scenario?
- Rigorously, subsections of the models are not well cited. Please make sure that all equations are properly cited.
- What is the nature of the inter-NOMA cell interference in SINR formulation?
- With regard to the problem formulation section, a brief explanation about the goal of the proposed method and its verification is require.
- The sub-channel allocation algorithm itself needs to be illustrate more briefly.
- The path loss model for both D2D and cellular is same. Please explain.
Without comparing the performance of the proposed method with benchmarks, how the authors have verified that their method has novelty. Overall the results are not compelling, it is encouraged to compare the performance of the proposed method with benchmarks.
Reviewer 2 Report
In this paper the authors investigate full-duplex cooperative NOMA systems with underlayed D2D communication. The authors have also validated their cooperative relaying and matching theory based resource allocation methods via extensive simulations. The paper's technical contributions are suitable for journal publications. Moreover, the authors have shown good organization structure and nice presentation of results. A good summary of the contributions and their difference and missing items with respect to existing state of art are also provided. Therefore I recommend the paper's acceptance.
Some minor comments:
- Some sentences can be reduced in the text. For example the last sentence of Conclusion section is too long.
- A notation table after introduction section will be helpful.
- In Section 2 there is only one subsection (section 2.1) which need to be merged as there is no Section 2.2.
- A summary of main observations of simulation results as well as trade off analysis of the proposed algorithm can be provided before Conclusions section.
Round 2
Reviewer 1 Report
Accept in Present Form.